# Photochemical permutation of *meta*-substituted phenols

Maialen Alonso [1,4], Giovanni Lonardi[1,4], Enrique M. Arpa[1], Baptiste Roure[1,2], Alessandro Ruffoni [3] ✉ & Daniele Leonori [1] ✉

Phenols and their derivatives are highly relevant motifs in pharmaceuticals, natural products, and other functional materials. Conventional strategies for phenol synthesis rely on classical aromatic functionalization, which is often dictated by electronic and steric factors. Herein, we report an alternative approach for phenol synthesis where irradiation in the presence of Lewis or Brønsted acids enables the selective migration of alkyl and aryl groups from *meta* to either the *ortho* or *para* positions. This method exploits the intrinsic photochemical properties of phenolic arenium ions and their rearrangement via 4π electrocyclization and following "1,2-methylene shift". By leveraging selective photoexcitation of these species, we can achieve precise control over the directionality of the permutation process. Specifically, short-wavelength irradiation (λ = 310 nm) promotes *meta→para* migration, while longer-wavelength irradiation (λ = 390 nm) *meta→ortho*. This approach offers a late-stage method to use readily available phenols as templates for the preparation of other isomers without de novo synthesis. The applicability of the method has been demonstrated on the isomerization of poly-substituted derivatives including some bioactive species.

Phenols and their derivatives are molecules of high societal relevance (Fig. 1a)[1–3]. For instance, 62% of small-molecule drugs approved by the FDA in 2020 contain phenol or phenolic ether unit[2]. These species include bioactive compounds like morphine[4], the gold standard for severe pain management, and tetrahydrocannabinol (THC)[5], which has evolved from recreational use to a therapeutic agent for pain relief and epilepsy treatment. Phenols are also prevalent in natural products and secondary metabolites, such as capsaicin, responsible for food spiciness, and resveratrol, an antioxidant with strong implications in anti-aging research[2].

The substitution pattern of phenols is crucial to their biological activities as it influences factors like O–H acidity and redox profile, which control their ability to participate in H-bonding and π,π interactions with biological systems[6–8]. Hence, when developing bioactive leads, it is crucial to access and evaluate derivatives with substituents at different aromatic positions to understand their

relevance and impact on interactions within the biological space[9]. This endeavor is currently approached through individual de novo synthesis of each derivative type (Fig. 1b). However, this can be often challenging, as phenol synthesis must follow the rules of aromatic chemistry, such as S$_E$Ar (electrophilic aromatic substitution) for phenol functionalization, which is mostly controlled by electronics (*ortho*, *para* directing)[7], or C($sp^2$)−H borylation followed by oxidation for aromatic oxygenation, which is generally governed by sterics (*meta*)[10,11]. Derivatives with substitution patterns that challenge these reactivity rules can be difficult to prepare, often requiring multistep synthetic sequences. The development of processes able to use readily available phenols as templates for the preparation of other isomers might greatly simplify these synthetic efforts and provide concise access to high-value materials[12].

Despite their synthetic potential, the development of processes moving substituents on aromatic rings are inherently challenging as

[1]Institute of Organic Chemistry, RWTH Aachen University, Aachen, Germany. [2]Department of Chemistry, University of Manchester, Manchester, UK. [3]Otto Diels—Institute of Organic Chemistry, Christian Albrecht Universitat zu Kiel, Kiel, Germany. [4]These authors contributed equally: Maialen Alonso, Giovanni Lonardi. ✉e-mail: aruffoni@oc.uni-kiel.de; daniele.leonori@rwth-aachen.de

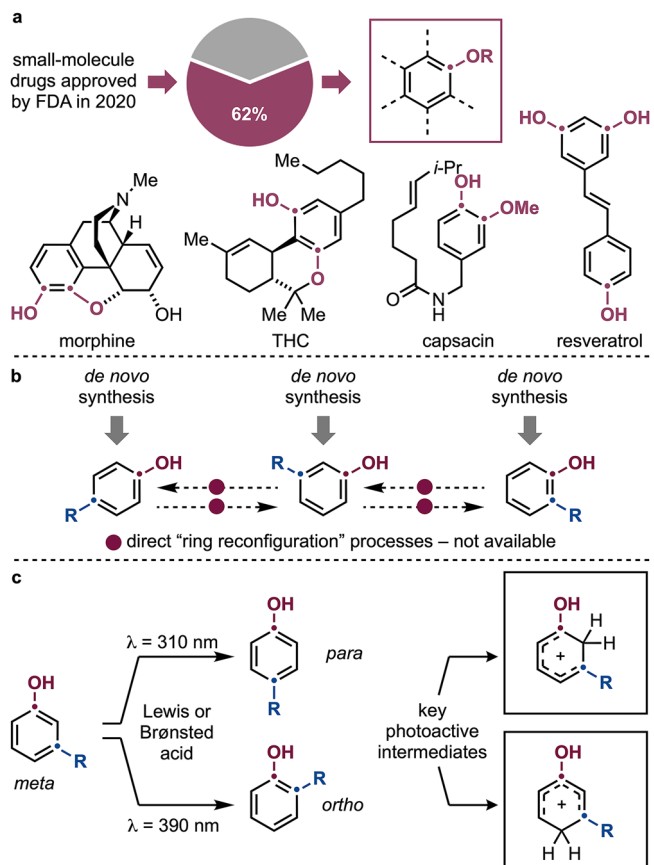

**a** small-molecule drugs approved by FDA in 2020

62%

morphine · THC · capsacin · resveratrol

**b** *de novo* synthesis · *de novo* synthesis · *de novo* synthesis

● direct "ring reconfiguration" processes – not available

**c** λ = 310 nm · Lewis or Brønsted acid · λ = 390 nm · *para* · *ortho* · key photoactive intermediates · *meta*

**Fig. 1 | Relevance of phenols in medicinal chemistry, their synthesis and our reaction design. a** Phenols are high-value materials encountered in the structure of natural products and blockbuster drugs. **b** The possibility to reconfigure the substitution patterns of phenols might streamline their preparation. **c** This work demonstrates a permutation concept that selectively shifts *meta*-substituents to either the *ortho* or the *para* positions.

## Results

At the outset of our work, we were inspired by Childs' pioneering studies on the structural interconversion of phenols under irradiation in the presence of superacid solvents or Lewis acids[18–20]. Under these conditions, phenols underwent dearomatization via protonation and requires reactivity patterns different to the ones adopted for aromatic functionalization. As an example, Lumb has recently introduced an oxygen group ring-walk strategy applicable to *para*-substituted phenols[13]. This method requires three steps and starts with the conversion of the phenol into a diazonium intermediate for selective 1,2-oxygen shift.

We and others recently became interested in developing photochemical strategies for chemical permutation, where fully functionalized (hetero)aromatics, such as thiazoles, isothiazoles, and indazoles, serve as starting materials for generating isomeric derivatives via substituent or ring-atom migration[14–17]. Within this framework, the ability to rearrange phenols would offer a streamlined route to high-value derivatives. Herein, we have studied the photochemical permutation of phenols in the presence of Lewis or Brønsted acids (Fig. 1c). This method leverages the distinct photochemical properties of phenolic arenium ions and provides selective directionality to the migration of alkyl and aryl groups from *meta* to *ortho* or *para* positions. The isomerization of substituents over the aromatic core enables the use of substituted phenols as templates for the preparation of isomeric derivatives without de novo synthesis.

ring-contraction into lumiketone intermediates, which subsequently rearrange[21–24]. Despite the potential for late-stage phenol modification, this reactivity has been explored only with a limited set of methylated derivatives and led to mixtures of isomeric products. To provide synthetic utility of this concept, two main challenges need to be addressed. Firstly, the permutation directionality needs to be achieved and understood to ensure selective outcomes. Second, it would be desirable to establish experimental guidelines predicting the most likely isomeric products. However, analysis of this reactivity reveals a complex network of potentially interconverting intermediates, making directional control challenging to approach.

Using Me-substituted phenols as models, the three isomers, $2_1$ (*ortho*), $2_2$ (*meta*), and $2_3$ (*para*), can undergo protonation to form a series of isomeric arenium ions (**A**) (Fig. 2). Among these, the six more stabilized species **A1**–**A6** can be photoexcited, leading to 4π electrocyclization and the bicyclic cation intermediates (**B**). These species can further interconvert via "1,2-methylene shift"[25], generating different areniums and ultimately isomeric phenols (Fig. 2 and Supplementary Information Section 9). For instance, *para*-protonation of $2_2$ can lead to **A1**, which subsequently forms **B1**. This species can potentially interconvert into **B2**–**B5** through a cyclic network. Notably, while **B1**, **B2** and **B3** can revert to $2_2$, **B4** and **B5** can ring-open to **A4** and **A8**, respectively, both resulting in $2_3$ where the *meta*-Me group has migrated to the *para* position. A key conclusion from our initial hypothesis is that $2_1$ & $2_2$, as well as $2_2$ & $2_3$, can potentially interconvert, whereas $2_1$ & $2_3$ are not connected by any direct isomerization pathway (see below and Fig. 3 for a more detailed mechanistic description). This suggests that [*ortho*⇌*meta*] and [*para*⇌*meta*] permutations might be feasible, whereas direct [*ortho*⇌*para*] interconversion is not, but might still take place as part of a stepwise process.

Despite this mechanistic complexity, we hypothesized that directional control could be achieved by exploiting a series of factors like the differential photostability and photoreactivity of the various arenium intermediates **A** as well the ground-state properties of the different bicyclic cations **B**. While there is no prior direct knowledge on these photophysical as well as ground-state aspects, we hoped that fine-tune reaction conditions, such as solvent, additives, and light sources, might be used to exert precise control over the permutation process.

We initially evaluated *meta*-Me-phenol $2_2$ to explore its potential conversion into the other constitutional isomers $2_1$ (*ortho*) and $2_3$ (*para*) (Table 1). Reactions were conducted in $CH_2Cl_2$ at room temperature using various Lewis and Brønsted acids under different light sources. Irradiation at $\lambda = 310$ nm with 0.5 equiv. $AlBr_3$ under diluted conditions (c = 0.05 M) selectively yielded $2_3$ in 68% yield, accompanied by a minor amount of recovered $2_2$ (8%), with no formation of $2_1$ (entry 1). This outcome indicates complete and irreversible selectivity for the net migration of the *meta*-Me group to the *para* position. Interestingly, running this reaction at higher concentration (c = 0.1 M) resulted in lower efficiency, with substantial recovery of $2_2$ (entry 2). The use of strong Brønsted acids, e.g. TfOH (1.0 equiv.), was also explored. Although the reaction proceeded, it resulted in lower efficiency (42% yield) (entry 3). Switching the irradiation source to lower-energy light ($\lambda = 390$ nm) but at higher concentration (c = 0.1 M) dramatically altered the reactivity. Under these conditions, the formation of $2_3$ was suppressed, while $2_1$ was obtained in good yield (64%) (entry 4). This result provides a fully selective process for the *meta*→*ortho* isomerization. Similarly, the use of TfOH under these conditions led to the same outcome, albeit with a lower yield (27%) (entry 5). Control experiments confirmed that no reaction occurred in the absence of either light or Lewis/Brønsted acid, regardless of the wavelength ($\lambda = 310$ or 390 nm) (Supplementary Information Section 4.2).

Notably, any attempt to subject phenols $2_1$ and $2_3$ to photochemical permutation resulted in complete recovery of starting

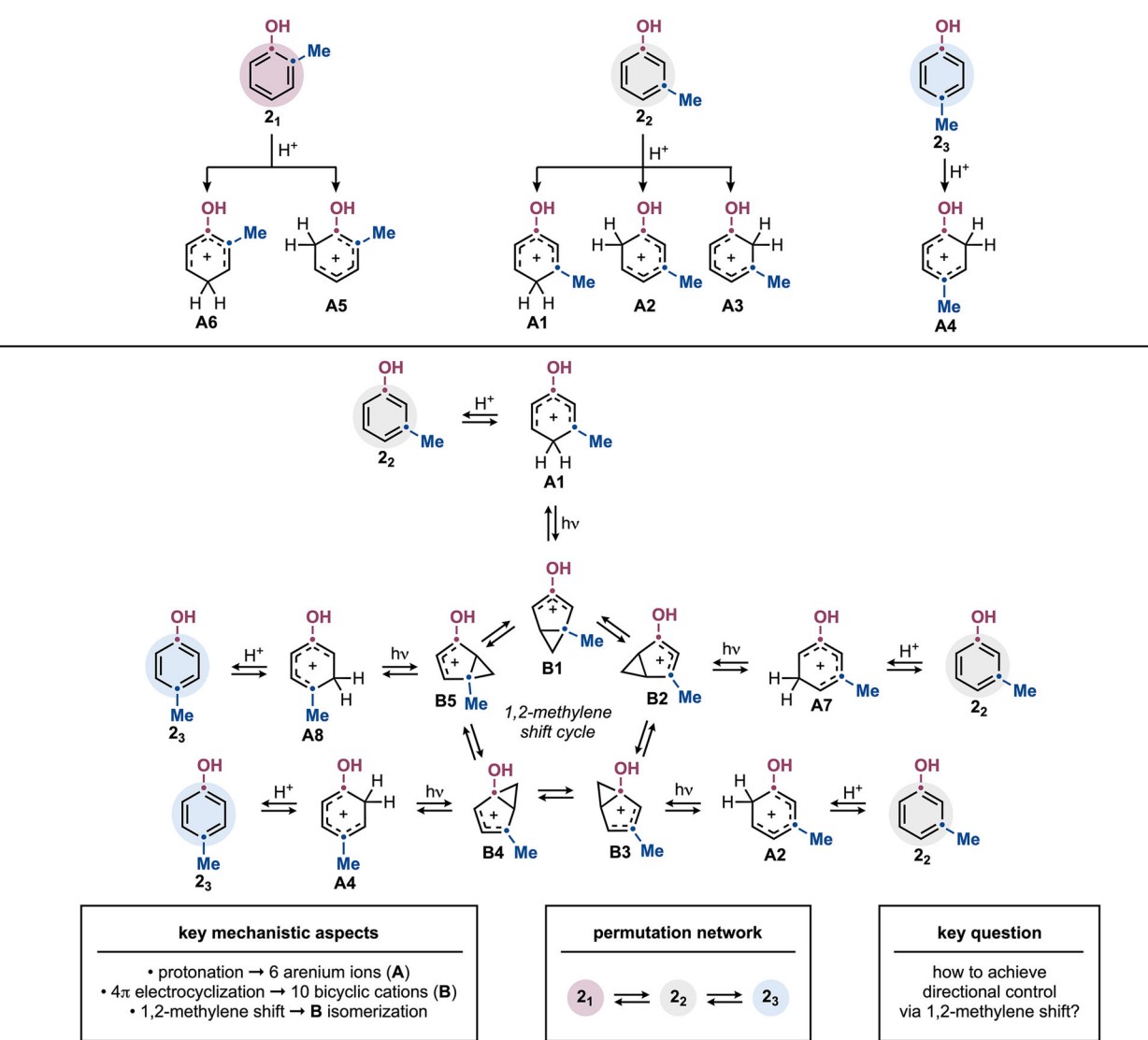

**Fig. 2 | Partial mechanistic picture and key challenges in phenol permutation.** Protonation of $2_1$–$2_3$ leads to arenium ions **A1**–**A6** that can isomerize via photochemical 4π electrocyclization.

material (Supplementary Information Section 4.2). This observation demonstrates that the permutation network is composed only by *meta→para* and *meta→ortho* isomerizations and raises fundamental questions of photochemical reactivity differentiating $2_2$ by $2_1$ & $2_3$ (Fig. 3a).

We aimed to elucidate the mechanism governing phenol permutations and to understand the wavelength-dependent switch in directional selectivity. Since the key photochemical step is the arenium 4π electrocyclization (**A→B**), and different arenium ions might exhibit distinct absorption profiles[20], the irradiation used can potentially enable selective photoexcitation.

UV/Vis absorption spectroscopy studies on $2_2$ revealed that the neutral species does not absorb in the 300–400 nm range (Fig. 3b). However, upon addition of TfOH, two distinct absorption bands emerged, one centered at $\lambda = 315$ nm and another at $\lambda = 350$ nm, with a tail extending into the blue region. If these bands correspond to different arenium ions, then selective photoexcitation might be achievable thus translating into distinct reaction pathways.

To explore this possibility, we considered the three most likely protonation products of $2_2$, **A1**–**A3**. TD-DFT studies revealed that these species exhibit markedly different $\lambda_{max}$ values, with **A1** (C4 protonation) being the most red-shifted. These computed absorption profiles

align well with the experimental UV/Vis data, reinforcing the hypothesis that wavelength-selective photoexcitation plays a key role in dictating reaction selectivity.

Accordingly, irradiation at $\lambda = 310$ nm should selectively excite **A1** over **A2** and **A3**, populating its (π,π*)-$S_1$ state. This excited state rapidly evolves through a barrierless pathway to a $S_1/S_0$ conical intersection with the ground state. At this intersection, internal conversion may either regenerate **A1** or trigger a 4π electrocyclization, forming **B1** (Fig. 3c). This high-energy intermediate can subsequently interconvert with **B2**–**B5** via a "1,2-methylene shift". Finally, ring-opening and re-aromatization ought to lead to the isomeric phenol $2_3$ (see the Supplementary Information Section 9 for a detailed mechanistic analysis). Alternatively, deprotonation of **B1/5** could lead to lumiketone intermediates (not shown) that are known to undergo photochemical ring opening upon population of its triplet state[22,23]. However, we have not been able to detect the formation of these species during the evaluation of $2_1$–$2_3$ as well as the substrates discussed below.

Notably, selective photoexcitation of **A1** ($\lambda = 310$ nm) and **A3** ($\lambda = 390$ nm) leads to different bicyclic cations, **B1** and **B6**, which belong to distinct "1,2-methylene shift" cycles. A key mechanistic consequence is that interconversions through **B1** can only regenerate $2_2$ or yield the *para* isomers $2_3$. Species **B7**–**B10**, arising from the

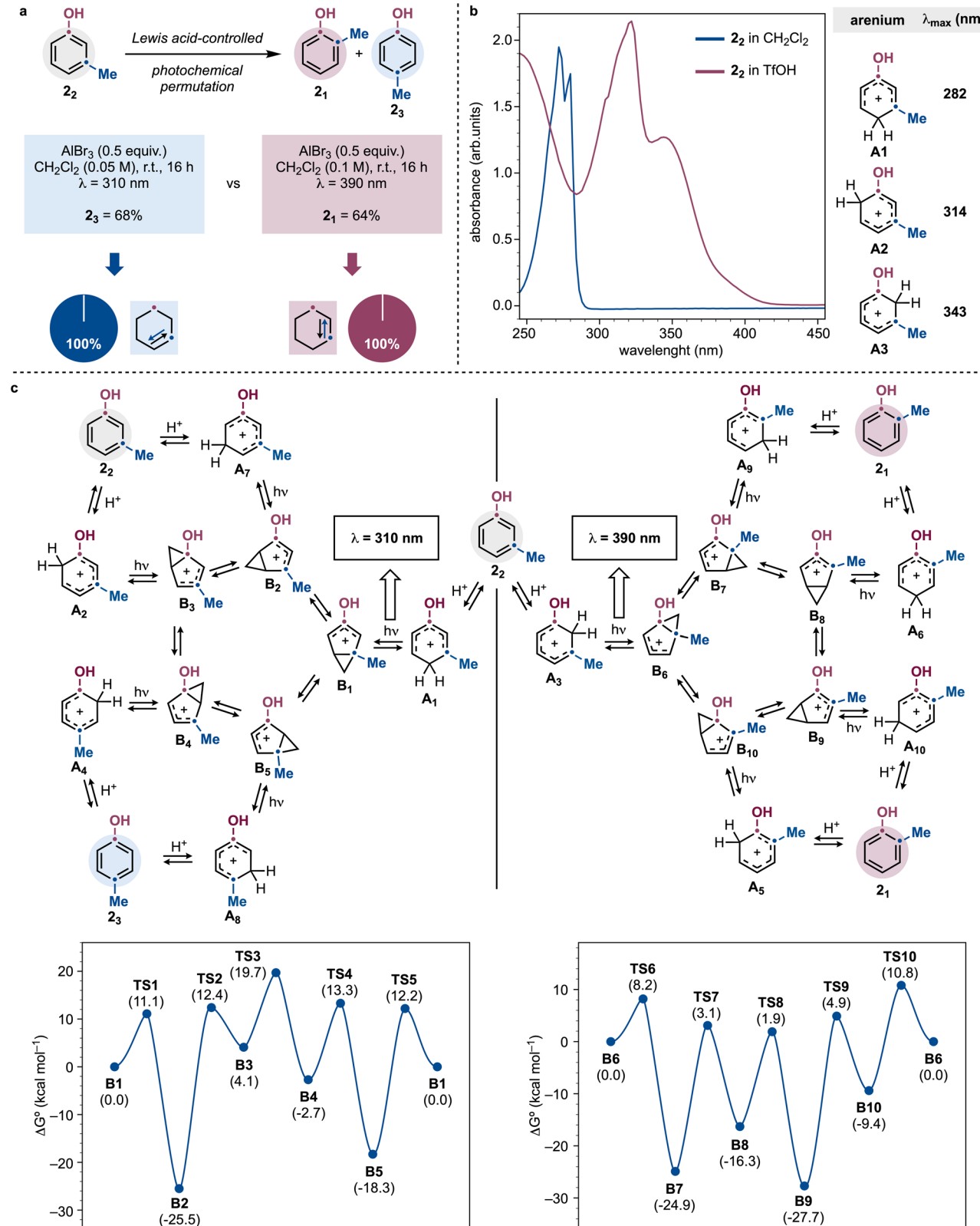

**Fig. 3 | Optimization and mechanistic insights of the reaction. a** Wavelength-dependent *meta→ortho* and *meta→para* isomerizations of **2₂**. **b** UV/Vis spectroscopy and TD-CAM-B3LYP/aug-cc-pVTZ/SMD(DCM)//CAM-B3LYP/cc-pVDZ studies on the protonation of **2₂** suggest the formation of arenium ions **A1–A3** that have different photophysical properties. **c** TD-CAM-B3LYP/aug-cc-pVTZ/SMD(DCM)//CAM-B3LYP/cc-pVDZ mechanistic analysis of the pathways leading to *meta→ortho* and *meta→para* isomerization of **2₂** into **2₁** and **2₃** and the energetic aspects for the isomerization of the bicyclic intermediates **B1–B5** and **B6–B10**.

**Table 1 | Optimization for the *meta→ortho* and *meta→para* isomerizations of $2_2$**

| Entry | λ (nm) | Acid | c (M) | $2_1$ (%) | $2_3$ (%) | rsm (%) |
|---|---|---|---|---|---|---|
| 1 | 310 | $AlBr_3$ (0.5 equiv.) | 0.05 | – | 68 | 8 |
| 2 | 310 | $AlBr_3$ (0.5 equiv.) | 0.1 | – | 20 | 53 |
| 3 | 310 | TfOH (2 equiv.) | 0.05 | – | 42 | 32 |
| 4 | 390 | $AlBr_3$ (0.5 equiv.) | 0.1 | 64 | – | 15 |
| 5 | 390 | TfOH (2 equiv.) | 0.1 | 27 | – | 31 |

[1]H NMR yields are reported.

interconversion of **B6**, exclusively result in methyl migration to the *ortho*-position and phenol $2_1$. This correctly explains the directional selectivity observed upon changing the irradiation sources.

We propose that the observed selectivity, as well as the lack of reactivity for $2_1$ and $2_3$, can be explained by considering the energetic factors controlling the isomerization of **B1–B5** and **B6–B10**. For instance, **B1** may either revert to **A1** or isomerize to **B2** or **B5**. Both pathways (i.e. **B1→B2** and **B1→B5**) are highly exothermic and have comparable energy barriers. However, further interconversion (e.g., **B2→B3**) are unlikely due to high barriers. Consequently, ring-opening and re-aromatization of **B2** and **B5** regenerate $2_2$ or lead to the formation of *para* isomer $2_3$. Interestingly, C6 protonation of $2_2$ yields arenium ion **A2**, which our TD-DFT calculations suggest may also be excited at $\lambda = 310$ nm (Fig. 3b). However, its corresponding ring-contracted product, **B3**, would face large barriers for further isomerization. Thus, we propose that the observed *meta→para* reactivity is the result of C4 protonation and selective photoexcitation of **A1**.

The photochemical stability of $2_3$ might also be a result of this energetic scenario. Indeed, *ortho*-protonation to **A4** and 4π electrocyclization and isomerization to **B5** should only allow for reversion back to $2_3$ due to high kinetic barriers preventing further isomerization (i.e., **B5→B1** or **B4→B3**).

A similar energetic profile applies to the **B6–B10** interconversion, which selectively leads to the formation of $2_1$. Specifically, C2 protonation of $2_2$ to **A3** and photoexcitation at $\lambda = 390$ nm gives **B6**, which then follows low-barrier isomerization pathways (**B6→B7** and **B6→B10**) to ultimately yield $2_1$.

Overall, while each arenium ion may undergo photoexcitation, the energetic landscape of the resulting bicyclic intermediates can govern the reactivity resulting in the observed *meta→ortho* and *meta→para* isomerizations.

A crucial mechanistic consequence of the isomerization mechanism discussed above is that, while several pathways in the $2_1$–$2_3$ network (see Fig. 3c) lead to the same product (e.g. **B4** and **B5** form $2_3$), the phenols ring carbons undergo permutation. For polysubstituted derivatives, this inevitably results in multiple products arising from the same system of interconverting intermediates. Indeed, considering the six dimethyl-containing phenols ($1_1$–$1_6$), full analysis of the permutation mechanism reveals the potential for ten arenium ions interconverting via twenty bicyclic cationic intermediates (see Supplementary Information Section 10 for more details) (Fig. 4a).

From this analysis, we identified a permutation system based on six potentially reversible isomerizations ($1_1 \leftrightarrows 1_2$, $1_1 \leftrightarrows 1_4$, $1_1 \leftrightarrows 1_5$, $1_2 \leftrightarrows 1_3$, $1_4 \leftrightarrows 1_5$, and $1_5 \leftrightarrows 1_6$) and two irreversible processes ($1_2 \rightarrow 1_4$ and $1_3 \rightarrow 1_4$). Thus, we were particularly interested in evaluating the behavior of the

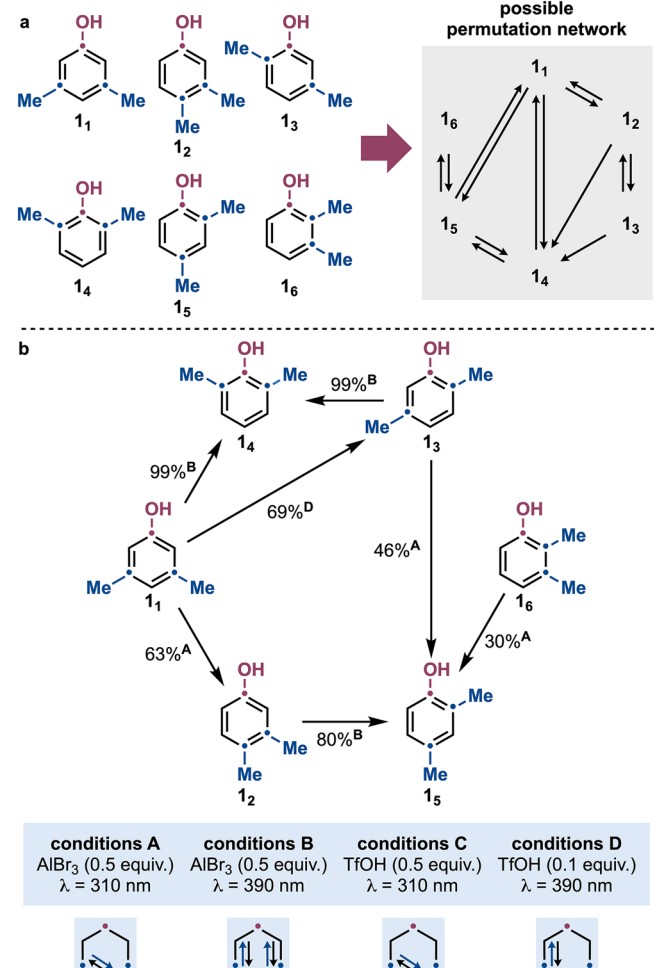

**Fig. 4 | Overview of the directionality control achieved over the permutation network. a** Potential permutation network on dimethyl-substituted phenols $1_1$–$1_6$. **b** Permutation network developed experimentally across $1_1$–$1_6$.

six dimethylated phenols $1_1$–$1_6$ under photochemical conditions and understand if selective isomerization might be achieved by switching irradiation wavelengths and fine tuning of other reaction parameters.

Starting with the symmetrical *meta,meta*-derivative $1_1$, irradiation at $\lambda = 310$ nm with 0.5 equiv. $AlBr_3$ (conditions A) selectively

**Fig. 5 | Substrate scope for the *meta→ortho* and *meta→para* isomerization of phenols.** Isolated yields are reported (see Supplementary Information Section 7 for details on recovered starting material for each scope entry) and TD-CAM-B3LYP/ aug-cc-pVTZ/SMD(DCM)//CAM-B3LYP/cc-pVDZ maximum absorption wavelengths for **A11**–**A13**. *λ = 370 nm. #λ = 350 nm. PFP *para*-fluorophenyl.

yielded 3,4-dimethylphenol $1_2$ in 63%, along with minor amounts of 2,5-dimethylphenol $1_3$ (11%) and recovered $1_1$ (12%), indicating full selectivity for the net movement of one *meta*-Me to the *para* position (Table 2). Notably, no formation of $1_4$, $1_5$, or $1_6$ was observed. Also in this case, switching to λ = 390 nm irradiation (conditions B) led to a shift in reactivity: formation of $1_2$ and $1_3$ was suppressed, while $1_4$ was obtained in quantitative yield. This fully selective process features the migration of both *meta*-Me groups to the *ortho* positions. Interestingly, this permutation outcome is not directly predicted by our mechanistic model (Fig. 4a). However, a stepwise pathway via $1_3$ could provide a plausible explanation, providing that $1_3$ exhibits similar photophysical and photochemical properties, allowing for a tandem isomerization. To test this hypothesis, we subjected $1_3$ to conditions B, yielding $1_4$ also in quantitative yield. Kinetic analysis of the conversion of $1_1$ to $1_4$ revealed that $1_3$ accumulates to a maximum concentration at 20 min, after which it decays into $1_4$ (Supplementary Information Section 4.1).

Further screening with Brønsted acids at λ = 310 nm revealed that TfOH afforded $1_2$ in 50% yield, with 24% unreacted $1_1$ (conditions C). Notably, using TfOH and irradiating the reaction at λ = 390 nm triggered an additional switch, this time favoring $1_3$ (59%) over $1_2$ or $1_4$. Here, the overall transformation corresponds to a selective migration of one *meta*-Me group to the *ortho* position. Furthermore, reducing TfOH to 0.1 equiv. further improved the yield of $1_3$ to 69% (conditions D). This result demonstrates how changes in reaction conditions (TfOH vs. AlBr$_3$) can have a dramatic impact on the photoreactivity of phenols as well as their arenium and bicyclic cation intermediates.

**Table 2 | Optimization for the *meta→ortho* and *meta→para* permutations on $1_1$**

| Entry | Conditions | $1_1$ (%) | $1_2$ (%) | $1_3$ (%) | $1_4$ (%) | $1_5$ (%) | $1_6$ (%) |
|---|---|---|---|---|---|---|---|
| 1 | A | 12 | 63 | 11 | – | – | – |
| 2 | B | – | – | – | 90 | – | – |
| 3 | C | 24 | 50 | – | – | – | – |
| 4 | D | – | – | 69 | – | – | – |

Conditions: A = AlBr$_3$ (0.5 equiv.), $\lambda$ = 310 nm; B = AlBr$_3$ (0.5 equiv.), $\lambda$ = 390 nm; C = TfOH (0.5 equiv.), $\lambda$ = 310 nm; D = TfOH (0.5 equiv.), $\lambda$ = 310 nm.
$^1$H NMR yields are reported.

To gain further insight, we synthesized $1_2$–$1_6$ and subjected them to analogous photochemical conditions. Irradiation of $1_2$ under conditions B yielded $1_5$, mirroring the *ortho* migration observed for $1_1$. However, under conditions A, C, and D, $1_2$ remained unreactive. Similarly, both $1_3$ and $1_6$ underwent conversion to $1_5$ under conditions A, while $1_3$ quantitatively formed $1_4$ under conditions B. This resulted in the permutation network depicted in Fig. 4b, where dimethylphenols $1_4$ and $1_5$ are photostable, while $1_3$ is photoreactive but can be selectively accumulated by fine-tuning reaction parameters.

Overall, this screening of photochemical conditions over substrates $2_1$–$2_3$ and $1_1$–$1_6$ provided a general framework of reactivity patterns:

(1) Near-visible light ($\lambda$ = 370–390 nm) promotes *ortho* migration of *meta*-substituents.

(2) Shorter wavelengths ($\lambda$ = 310 nm) favor *para* migration.

(3) Substituents already positioned in *ortho*, or *para* do not undergo migration under any tested conditions.

To demonstrate the synthetic utility of this permutation strategy, we explored a range of phenols featuring various substitution patterns (Fig. 5). Each substrate was evaluated under conditions A–D to accommodate differences in absorption properties of the corresponding arenium ions and facilitate the migration of *meta*-substituents to either the *ortho*- or *para*-positions.

We began by examining *meta*-aryl derivatives ($3_1$–$5_1$), which underwent efficient *para*-rearrangement ($3_2$–$5_2$) upon irradiation with AlBr$_3$ at $\lambda$ = 390 nm. Interestingly, this photochemical behavior differed from that observed for compounds $2_2$ and $1_2$ (see above), where $\lambda$ = 390 nm irradiation led to *meta→ortho* rearrangement of methyl groups. UV/Vis absorption spectroscopy and TD-DFT studies on areniums A11–A13 revealed significantly red-shifted $\lambda_{max}$ values. Specifically, A11, the intermediate leading to *ortho*-products, was determined to primarily absorb in the blue region and therefore we propose it might not be excited by the purple LEDs used in this study. Attempts to induce rearrangement with blue LEDs were unsuccessful (see Supplementary Information Section 4.3 for more details).

We next investigated unsymmetrical *meta*-Me/*meta*-substituted phenols ($6_1$–$9_1$), which selectively yielded *meta*-*ortho* derivatives ($7_2$–$9_2$) in moderate to good yields under conditions B. Notably, selective formation of $6_1$ was achieved, by migration of both Me and Et substituents. In this case, reaction monitoring suggests the Me-shift is occurring prior the Et-shift, which might be due to steric effects (see Supplementary Information Section 4.5). These reactions demonstrated tolerance for *meta*-alkyl ($6_1$), aryl ($7_1$), F ($8_1$), and OMe ($9_1$) groups, while enabling selective Me-substituent rearrangement.

Attempts to generate *meta*-*para* derivatives were unsuccessful, likely due to steric hindrance in the bicyclic intermediates generated in the process.

Substrates with 2,5-disubstitution ($10_1$–$12_1$) were then evaluated. Under conditions B, we successfully promoted the *meta→ortho* migration of the 5-Me group, affording 2,6-disubstituted derivatives ($10_3$, $11_2$ and $12_2$). In the case of $10_1$, we also achieved *meta→para* rearrangement to $10_2$, albeit in lower yield.

Given the prevalence of the 2,5-disubstitution pattern in natural phenols, we applied our conditions to naturally occurring derivatives. Carvacrol ($13_1$), known for its antibacterial properties[26], was selectively converted into either isocarvacrol ($13_2$, Conditions A) or the antiseptic *o*-thymol ($13_3$, Conditions B) in good yields[26], demonstrating that the bulkier *i*-Pr group participates in the reactivity. Similarly, antibacterial thymol[26,27] ($14_1$) was transformed into either *o*-thymol ($13_3$) or isothymol ($14_2$), while the anti-infective amylmetacresol[28] ($15_1$) yielded $15_2$ and $15_3$ in good to high yields.

Phenols with 2,3- ($18_1$) and 3,4-disubstitution ($16_1$–$17_1$) underwent a single isomerization pathway, forming 2,4-products via *meta→para* ($18_2$) and *meta→ortho* ($16_2$–$17_2$) isomerizations, respectively.

We further explored trisubstituted phenols. The 2-Cl-4,5-(Me)$_2$ derivative ($19_1$) underwent *meta→ortho* migration in excellent yield. The antiseptic drug chloroxylenol[29] ($21_1$) afforded a mixture of 2,4,5-(80%) and 2,4,6-(11%) derivatives ($21_2$ and $21_3$) under $\lambda$ = 370 nm irradiation. The 3,5-(Me)$_2$–4-Ph phenol ($20_1$) exhibited wavelength-dependent selectivity, undergoing single or double *meta→ortho* Me-migration upon irradiation at $\lambda$ = 350 or 390 nm, respectively. Finally, the 2,4,5-substituted derivative ($22_1$) selectively yielded either 2,4,5-($22_2$) or 2,3,6-($22_3$) phenols in moderate to excellent yield. Notably, $22_3$ could be further isomerized to the 2,4,6-derivative ($22_4$) by shifting the irradiation wavelength to 310 nm.

As a potential future direction, we obtained preliminary results suggesting that this strategy could be extended to phenolic ethers. Irradiation of *meta*-Me-anisole $23_1$ in the presence of TfOH resulted in selective isomerization, yielding rearranged products $23_2$ and $23_3$ in moderate to good yields and full selectivity.

In conclusion, we have developed a photochemical strategy for the selective isomerization of *meta*-phenols, which rearranges their structures with precise control over substitution patterns. This methodology exploits the distinct photophysical and electronic properties of protonated phenols to direct the migration of *meta*-substituents to either the *ortho* or *para* on the basis of the irradiation wavelength. Mechanistic investigations, informed by UV/Vis spectroscopy and TD-DFT calculations, revealed that selective photoexcitation of arenium ions dictates the reaction pathways, via initial 4π electrocyclization. The energetic

landscape of the resulting bicyclic cation intermediates further governs the observed isomerization patterns, ensuring that only *meta→ortho* and *meta→para* isomerizations occur. Overall, this approach provides an alternative to traditional phenol synthesis as substituents are not introduced in the aromatic core but rather moved. This enables the use of functionalized phenols as template for the preparation of structural isomers. Future efforts will focus on expanding the scope to other aromatics to further enhance synthetic utility.

## Methods

### General procedure for the photochemical permutation of *meta*-phenols

In an Argon filled glove box, a dry tube equipped with a stirring bar was charged with the corresponding phenol (0.1 mmol, 1.0 equiv.) followed by AlBr$_3$ (0.5 equiv.). Dry and degassed CH$_2$Cl$_2$ (0.1 M) was added and the tube was capped with a Supelco aluminum crimp seal with septum (PTFE/butyl). The tube was placed into a Helios photoreactor equipped with the 310 nm lamps and a fan or under a 390 nm Kessil lamp with a distance from the lamp to the bottom of the vial of 4 cm. The lamps and the fan were switched on and the mixture was stirred under irradiation for 16 h. The photoreactor and the fan were switched off. The mixture was diluted with H$_2$O (2 mL). The organic layer was separated and the aqueous layer was extracted with CH$_2$Cl$_2$ (2 mL x 2) and the combined organic layers were dried (MgSO$_4$), filtered and evaporated. The residue was purified by column chromatography on silica gel to give the desired product.

## Data availability

All data from optimization studies, experimental procedures, mechanistic studies and product characterization are available within the paper, Supplementary Information and Coordinate files. Additional data can be available from the corresponding authors upon request. Source data are provided with this paper.

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

## Acknowledgements

This work was supported by the European Research Council (101086901, D.L.) and the Marie Skłodowska-Curie Actions (101150093-BRADOCO, G.L. and 101150311-NOZONE, E.M.A.). The authors also thank Sanofi for financial support (M.A.) and Janssen for a PhD CASE Award (B.R.). The authors gratefully acknowledge the computing time provided to them at the NHR Center NHR4CES at RWTH Aachen University (project number p0021519). This is funded by the Federal Ministry of Education and Research, and the state governments participating on the basis of the resolutions of the GWK for national high-performance computing at universities (www.nhr-verein.de/unsere-partner).

## Author contributions

D.L. and A.R. designed the project. M.A., G.L. and B.R. ran all synthetic experiments; E.M.A. ran all the computational studies. All authors discussed the results and wrote the manuscript.

## Funding

## Competing interests

The authors declare no competing interests.
