## [Transparent Peer Review file · Nature Communications]

Photochemical Permutation of meta-Substituted Phenols

Corresponding Author: Professor Daniele Leonori

Version 0:

Reviewer comments:

Reviewer #1

(Remarks to the Author)

This is an excellent paper which reports a novel transformation. In terms of novelty, I have not seen a transformation like this before and I think the reaction and the mechanism are interesting. I don't think this paper will significantly impact medicinal chemistry, particularly as the starting material is somewhat esoteric. However, I think the authors deserve a lot of credit for this creative reaction, and I think synthetic chemists will find this photochemical rearrangements intriguing, and it is also a distinct approach in the molecular editing area. This is exactly what should be published in NC.

Minor points to address:

1. Clarity of schemes : We recommend annotating the electron transfer pathways in key intermediates of Figure 2, which would significantly enhance readers' comprehension of the reaction mechanism.
2. We recommend including a table summarizing unsuccessful examples. It would be more useful in the main manuscript. These results are important.
3. Why are most migrating groups methyl groups? Could examples of different alkyl migrations be included?
4. "Is it feasible to synthesize disubstituted phenols bearing two distinct meta-alkyl groups and employ intramolecular competition experiments to compare migration rates, thereby elucidating underlying trends?"

Reviewer #2

(Remarks to the Author)

I strongly support publication. I truly have minor considerations listed below. I find the concept novel, interesting and potentially impactful. Certainly room for future improvement to scope and yield, but the work sets a new direction from a conceptual perspective.

Minor Comments on the Manuscript

1. This is a stylistic comment, so I leave it to the authors to decide, but I have to be very honest that the nomenclature and extensive use of numbers made it very difficult for me to follow the writing and figures, especially in figures 3 and 4. I have some sympathy for the authors, in that they are trying to describe a very complicated reaction network, but in illustrating the complete pathway, they are also incorporating a lot of complexity that could possibly be sent to the SI. I don't want to overstep my boundaries as a referee and make stylistic recommendations, but as someone who very much enjoyed this piece of work, I find it important to convey the message: in its current form, it is not easy to read/interpret the figures.
2. On page 10 of 14, reference to the Supp Info should be more precise – it is difficult to find the information that is referenced.
3. Is there an equilibrium between the positively charged arenium ions and a neutral species in which phenol adds to the arenium to afford a neutral adduct? Would this kind of a scenario affect the absorbance spectrum?
4. RE: the speciation of AlBr₃ – could the authors comment on the possibility of forming "super" lewis acids under these conditions, involving two equivalents of AlBr₃? This is a minor question, but it relates to the speciation of AlBr₃ under the conditions.

Comments on the SI

1. In the SI on page 25/55, Fig. S4 the UV-Vis of 11 shows a negative absorption at ~275 nm. Could the authors provide a better spectrum, or explain why the spectrum was not normalized.
2. Check the inset legend of Fig. S5, which currently identifies both traces as taken in CH₂Cl₂.

Reviewer #3

(Remarks to the Author)

Conventional strategies for phenol synthesis rely on classical aromatic functionalization. Here, the author report an alternative approach by irradiation in the presence of Lewis or Brønsted acids for phenol synthesis, which enables the selective migration of alkyl and aryl groups from meta to either the ortho or para positions. The results show that short-wavelength irradiation ($\lambda = 310$ nm) promotes meta to para migration, while longer-wavelength irradiation ($\lambda = 390$ nm) meta to ortho. Nevertheless, this irradiation strategy has in fact been studied in fully functionalized (hetero)aromatics for generating isomeric derivatives via substituent or ring-atom migration and were inspired by Childs' pioneering studies. Therefore, it seems inappropriate to publish at Nat. Commun., and more importantly, there are some issues that need to be taken seriously.

1. In the main text, the author proposed that the key conclusion is that 21 & 22, as well as 22 & 23, can potentially interconvert, whereas 21 & 23 are not connected by any directly isomerization pathway. However, necessary theoretical or experimental supporting information are not reflected in Figure 2. Although the author mentions that "A detailed analysis of all pathways is provided in the Supplementary Information", the directivity described in this sentence is too general to quickly find appropriate answers in Supplementary Information. If necessary, advise the author to explain important content in detail, especially in the text to improve readability.

2. In the main text, the author proposed that "This suggests that [ortho meta] and [para meta] permutations might be feasible". But what is represented in the Figure 1C is that [meta ortho] and [meta para] permutations might be feasible, and these processes seem to be irreversible. We advise the author to examine it carefully or give an explanation.

3. In the Figure 3A, AlBr₃ (entry 1 and entry 4) shows good selectivity under both $\lambda = 310$ nm and $\lambda = 390$ nm. However, TfOH is used in the controlled experiment of UV/Vis spectroscopy in Figure 3B, which seems to be illogical. What is the result if used AlBr₃ in UV/Vis spectroscopy?

4. The author provides a detailed procedure for the reaction optimization of 3,5-dimethylphenol (11) and 3-phenylphenol (31) in Supplementary Information. However, the detailed procedure for the reaction optimization of Me-substituted phenols (22, discussed as a model in main text) is completely missing. It is recommended that the author supplement it in the Supplementary Information.

5. In Figure 3A, it is shown that the acid also has regulatory effect on selectivity, and even causes selectivity inversion (entry 1 vs entry 3 / entry 4 vs entry 5), but the authors do not seem to mention the role of acid.

6. The author mentions that "considering the six dimethyl-containing phenols (11-16), full analysis of the permutation mechanism reveals the potential for ten arenium ions interconverting via twenty bicyclic cationic intermediates (see Supplementary Information for more details)" in Figure 4A. However, we were unable to accurately locate the corresponding analysis in the supplementary information and therefore cannot understand this result. Please point out the specific location in the SI for easy reading and comprehension. Ambiguity for the specific location of supporting information has always existed throughout the main text.

7. "Figure 2B" mentioned in the main text has not been marked accordingly in Figure 2, please check it carefully. And the labeling method of "A" and "B" is easy to be confused with the "A/B" about arenium ions (A) and bicyclic cations (B).

Reviewer #4

(Remarks to the Author)

Version 1:

Reviewer comments:

Reviewer #1

(Remarks to the Author)

The authors propose a selective migration method for phenols employing light and Lewis or Brønsted acids. This novel approach provides innovative ideas for molecular editing. While previous studies have investigated photochemical reactions in phenols, this work not only systematically studies the scope of the reaction and expands its applications, but also optimizes the reaction conditions. Therefore, I believe this work is meaningful. The authors have provided detailed responses to the reviewers' comments. Therefore, I believe it can be accepted for publication.

One small question

1) I can appreciate that such reactions are quite fascinating, but the figures still appear very difficult to understand. If possible, I hope the authors can simplify them to clarify the relationships between the mechanisms, products, and substrates.

Reviewer #2

(Remarks to the Author)

The authors have revised the manuscript according to the referees' requests. I support publication of the revised manuscript.

Reviewer #3

(Remarks to the Author)

This revised manuscript addresses most of my concerns from the previous review; however, there is still one issue remaining due to understanding biases :

In entries 1 and 3 of Figure 3a, replacing TfOH (entry 3) with AlBr₃ (entry 1) increases the ratio of 23:22 from 1.3 (42:32) to 8.5 (68:8). In my opinion, while maintaining a wavelength of 310 nm, AlBr₃ plays a significant role in selective control. Similarly, in entries 4 and 5 (with a wavelength of 390 nm), replacing TfOH (entry 5) with AlBr₃ (entry 4) increases the ratio of 21:22 from 0.9 (27:31) to 4.3 (64:15). Could the author explain why AlBr₃ enhances selectivity?

Reviewer #4

(Remarks to the Author)

Reviewer #1:

This is an excellent paper which reports a novel transformation. In terms of novelty, I have not seen a transformation like this before and I think the reaction and the mechanism are interesting. I don't think this paper will significantly impact medicinal chemistry, particularly as the starting material is somewhat esoteric. However, I think the authors deserve a lot of credit for this creative reaction, and I think synthetic chemists will find this photochemical rearrangements intriguing, and it is also a distinct approach in the molecular editing area. This is exactly what should be published in NC.

We thank the reviewer for their positive evaluation of our work.

Minor points to address:

1. Clarity of schemes. We recommend annotating the electron transfer pathways in key intermediates of Figure 2, which would significantly enhance readers' comprehension of the reaction mechanism.

I did not act on this comment as I am not sure about the statement on electron transfer. Figure 2 deals with protonation and following structural rearrangement. Maybe the Reviewer could further explain how we can improve the Figure?

2. We recommend including a table summarizing unsuccessful examples. It would be more useful in the main manuscript. These results are important.

We added a table to the Supporting Information in Section 11: "Unsuccessful Substrates".

3. Why are most migrating groups methyl groups? Could examples of different alkyl migrations be included?

In our scope we demonstrate the migration of *i*-Pr (**13₁**), Et (**6₁**), Ph (**3₁**), *p*-Tol (**4₁**), *p*-F-Ph (PFP) (**5₁**) groups. We have run additional experiments using mono-substituted *m*-Et (**S1₁**) and *m*-*t*-Bu (**S2₁**) phenols. Since their reactivity mirrors the one of other systems we have included these examples in the revised SI in Section 7.1: "Additional substrates". I hope this is OK.

4. "Is it feasible to synthesize disubstituted phenols bearing two distinct meta-alkyl groups and employ intramolecular competition experiments to compare migration rates, thereby elucidating underlying trends?"

This is an interesting point. We have run additional experiments with **6₁** that leads to **6₂** where both Me and Et groups moved to the *ortho* positions. Under our optimized conditions (16 h) we only observe **6₂** as reported in the manuscript. However, reaction analysis after 2 h of irradiation revealed the formation of 19% of **A** and 21% of **6₂** and 15% of unreacted **6₁**. Crucially, we did not observe the formation of **B**. We believe this suggests that the Me group shift is faster than the Et group and this might be a result of sterics. We have added this information in the revised manuscript referring the reader to the SI Section 4.5: "Competition Experiments". A full kinetic analysis is rather challenging in this case as there are other unproductive pathways operating and we do not manage to get reproducible data from where kinetic constants can be extrapolated. Unfortunately, we do not have available an NMR probe equipped with light irradiation. I hope this is OK.

h (nm)	additive	solvent	time	6 ₂ (%)	A (%)	B (%)	6 ₁ (%)
390	AlBr ₃ (0.5 eq)	CH ₂ Cl ₂ (0.1 M)	2 h	21	19	-	15
390	AlBr ₃ (0.5 eq)	CH ₂ Cl ₂ (0.1 M)	16 h	29	-	-	-

Reviewer #2

I strongly support publication. I truly have minor considerations listed below. I find the concept novel, interesting and potentially impactful. Certainly room for future improvement to scope and yield, but the work sets a new direction from a conceptual perspective.

We thank the reviewer for their positive evaluation of our work.

Minor Comments on the Manuscript:

1. This is a stylistic comment, so I leave it to the authors to decide, but I have to be very honest that the nomenclature and extensive use of numbers made it very difficult for me to follow the writing and figures, especially in figures 3 and 4. I have some sympathy for the authors, in that they are trying to describe a very complicated reaction network, but in illustrating the complete pathway, they are also incorporating a lot of complexity that could possibly be sent to the SI. I don't want to overstep my boundaries as a referee and make stylistic recommendations, but as someone who very much enjoyed this piece of work, I find it important to convey the message: in its current form, it is not easy to read/interpret the figures.

Thank you. I agree with the Reviewer that the schemes can be difficult to follow as they deal with a complex network of intermediates, which are all very similar. I would rather not move information from the schemes to the SI as I think the complex nature of this network of intermediates is one of the key aspects of the reactivity. I have revised Figure 2–4 showing all structures involved. I did not include the **B1–B10** in the computational setting as I felt it was too much. I hope this makes them easier to understand.

2. On page 10 of 14, reference to the Supp Info should be more precise – it is difficult to find the information that is referenced.

Thank you. We specified all the sections referencing the Supporting Information in the entire manuscript.

3. Is there an equilibrium between the positively charged arenium ions and a neutral species in which phenol adds to the arenium to afford a neutral adduct? Would this kind of a scenario affect the absorbance spectrum?

The reactivity proposed by the Reviewer would possibly lead to the formation of a diarylether, which we have never detected. Hence, we do not believe this to have a role in the reactivity discussed here.

4. RE: the speciation of AlBr_3 – could the authors comment on the possibility of forming "super" lewis acids under these conditions, involving two equivalents of AlBr_3 ? This is a minor question, but it relates to the speciation of AlBr_3 under the conditions.

We have explored computationally the feasibility for the formation of a superacid species in which one molecule of phenol coordinates two AlCl_3 molecules. The results show that coordination of the first AlCl_3 to the OH group is exergonic ($\Delta G^\circ = -9.8$ kcal/mol), while a second coordination, which we could only locate through the aromatic π -cloud, is highly endergonic ($\Delta\Delta G^\circ = +11.7$ kcal/mol). Albeit the overall dicomplexation is slightly endergonic, we exclude the possibility of forming dimeric species acting as superacid. I did not revise the manuscript and SI to add this aspect but would be happy to do so if the Reviewer and Editor deem it necessary.

Comments on the SI:

1. In the SI on page 25/55, Fig. S4 the UV-Vis of 11 shows a negative absorption at ~275 nm. Could the authors provide a better spectrum, or explain why the spectrum was not normalized.

Done.

2. Check the inset legend of Fig. S5, which currently identifies both traces as taken in CH₂Cl₂.

Done.

Reviewer #3

Conventional strategies for phenol synthesis rely on classical aromatic functionalization. Here, the author report an alternative approach by irradiation in the presence of Lewis or Brønsted acids for phenol synthesis, which enables the selective migration of alkyl and aryl groups from meta to either the ortho or para positions. The results show that short-wavelength irradiation ($\lambda = 310$ nm) promotes meta to para migration, while longer-wavelength irradiation ($\lambda = 390$ nm) meta to ortho. Nevertheless, this irradiation strategy has in fact been studied in fully functionalized (hetero)aromatics for generating isomeric derivatives via substituent or ring-atom migration and were inspired by Childs' pioneering studies. Therefore, it seems inappropriate to publish at Nat. Commun., and more importantly, there are some issues that need to be taken seriously.

I regret that our manuscript did not convince the Reviewers of the synthetic significance and novelty of our work. However, I hope the Reviewers and the Editor will consider the following points.

- While I fully acknowledge that the photochemistry of protonated phenols has been explored before by Childs (as we cited), I hope to believe that our work substantially advances this reactivity along multiple aspects which crucially involve the mechanistic pathways and the understanding of what can and cannot be achieved.
- These prior studies were limited to simple cresol derivatives and yielded unselective mixtures of isomers. In contrast, our study establishes wavelength-dependent control over the migration of alkyl and aryl substituents across a broad range of phenolic substrates, including poly-functionalized derivatives and bioactive examples.
- In our work we address the fundamental question: how is selectivity between *ortho* and *para* migration achieved in these photochemical rearrangements? To the best of our knowledge, this is the first attempt at tackling the mechanistic analysis of this transformation, and we hope the Reviewers and the Editor will recognize that this is far from trivial, given the intricate network of potential intermediates. A key conclusion of our work is that different arenium ions exhibit distinct photophysical properties, which can be selectively accessed through wavelength-controlled irradiation. This feature provides a powerful lever to direct and rationalize reactivity. Furthermore, the energy landscape and interconversion dynamics of the resulting bicyclic cations offer a solid framework to explain the observed selectivity. Taken together, I hope to believe this work represents a conceptual and mechanistic advance in the field of aromatic photochemistry.
- I am not sure about the claim that this chemistry has already been developed on "fully functionalized (hetero)aromatics". I have not found any other examples where heteroarenes featuring OH-functionalities undergo isomerization. We do have reported the permutation of (iso)thiazoles in a recent *Nature* paper. However, the mechanistic implications and substrate scopes do not share any similarities with what disclosed here.

1. In the main text, the author proposed that the key conclusion is that 21 & 22, as well as 22 & 23, can potentially interconvert, whereas 21 & 23 are not connected by any directly isomerization pathway. However, necessary theoretical or experimental supporting information are not reflected in Figure 2. Although the author mentions that "A detailed analysis of all pathways is provided in the Supplementary Information", the directivity described in this sentence is too general to quickly find appropriate answers in Supplementary Information. If necessary, advise the author to explain important content in detail, especially in the text to improve readability.

Thank you for raising this point. We discussed Figure 2 deliberately without too many details as we then delved into all details in Figure 3 where we wrote:

"...selective photoexcitation of A1 ($\lambda = 310$ nm) and A3 ($\lambda = 390$ nm) leads to different bicyclic cations, B1 and B6, which belong to distinct "1,2-methylene shift" cycles. A key mechanistic consequence is that

interconversions through **B1** can only regenerate **2₂** or yield the para isomers **2₃**. Species **B7–B10** – arising from the interconversion of **B6** – exclusively result in methyl migration to the ortho-position and phenol **2₁**. This correctly explains the directional selectivity observed upon changing the irradiation sources."

These two isomerization cycles explain why **2₁** & **2₂** and **2₂** & **2₃** can directly interconvert between each other, while **2₁** & **2₃** cannot.

Would it be OK if the text around Figure 2 directs the reader to following discussion and Figure 3? I am worried to repeat the same information in the manuscript. I have modified the text in this way:

"A key conclusion from this initial hypothesis is that **2₁** & **2₂**, as well as **2₂** & **2₃**, can potentially interconvert, whereas **2₁** & **2₃** are not connected by any direct isomerization pathway (see below and Figure 3 for a more detailed mechanistic description)"

Would this be OK?

2. In the main text, the author proposed that "This suggests that [ortho meta] and [para meta] permutations might be feasible". But what is represented in the Figure 1C is that [meta ortho] and [meta para] permutations might be feasible, and these processes seem to be irreversible. We advise the author to examine it carefully or give an explanation.

Sorry about this, the Reviewers are right. We have rectified the arrows in the text.

3. In the Figure 3A, AlBr₃ (entry1 and entry 4) shows good selectivity under both $\lambda = 310$ nm and $\lambda = 390$ nm. However, TfOH is used in the controlled experiment of UV/Vis spectroscopy in Figure 3B, which seems to be illogical. What is the result if used AlBr₃ in UV/Vis spectroscopy?

Thank you for raising this point. In Section 6 of the SI we wrote: "We used TfOH for the UV spectroscopy studies to fully shift the equilibrium to the protonated species **A1-A3**."

We have now revised the text adding the following: "These analyses were not conducted using AlBr₃ because the high dilution required for good resolution leads to decomplexation." I hope this is OK.

4. The author provides a detailed procedure for the reaction optimization of 3,5-dimethylphenol (11) and 3-phenylphenol (31) in Supplementary Information. However, the detailed procedure for the reaction optimization of Me-substituted phenols (22, discussed as a model in main text) is completely missing. It is recommended that the author supplement it in the Supplementary Information.

We have added this information to the revised Supplementary Information.

5. In Figure 3A, it is shown that the acid also has regulatory effect on selectivity, and even causes selectivity inversion (entry1 vs entry 3 / entry 4 vs entry 5), but the authors do not seem to mention the role of .acid.

I am not sure about this comment. The use of AlBr₃ vs. TfOH does not alter the selectivity of the process.

6. The author mentions that "considering the six dimethyl-containing phenols (11-16), full analysis of the permutation mechanism reveals the potential for ten arenium ions interconverting via twenty bicyclic cationic intermediates (see Supplementary Information for more details)" in Figure 4A. However, we were unable to accurately locate the corresponding analysis in the supplementary information and therefore cannot understand this result. Please point out the specific location in the SI for easy reading and comprehension. Ambiguity for the specific location of supporting information has always existed throughout the main text.

Thank you for raising this point. This information should have been in the SI, sorry about that. We have now added it (Section 10).

7. "Figure 2B" mentioned in the main text has not been marked accordingly in Figure 2, please check it carefully. And the labeling method of "A" and "B" is easy to be confused with the "A/B" about arenium ions (A) and bicyclic cations (B).

We added the section to Figure 2 and in general we changed the names of the sections of all the figures with lowercase letters (a,b,c) to avoid confusion. Thank you.

Reviewer #4

Reviewer #1:

The authors propose a selective migration method for phenols employing light and Lewis or Brønsted acids. This novel approach provides innovative ideas for molecular editing. While previous studies have investigated photochemical reactions in phenols, this work not only systematically studies the scope of the reaction and expands its applications, but also optimizes the reaction conditions. Therefore, I believe this work is meaningful. The authors have provided detailed responses to the reviewers' comments. Therefore, I believe it can be accepted for publication.

We thank the reviewer for their positive evaluation of our work.

One small question

1) I can appreciate that such reactions are quite fascinating, but the figures still appear very difficult to understand. If possible, I hope the authors can simplify them to clarify the relationships between the mechanisms, products, and substrates.

I really don't know what to do to further simplify the schemes. The mechanistic picture is indeed very complex. However, in my opinion it is crucial to have it in the manuscript. I have tried to change colors to the various intermediates but in my opinion the schemes worsen. I propose to leave them as they are.

Reviewer #2:

The authors have revised the manuscript according to the referees' requests. I support publication of the revised manuscript.

We thank the reviewer for their positive evaluation of our work.

Reviewer #3:

This revised manuscript addresses most of my concerns from the previous review; however, there is still one issue remaining due to understanding biases:

In entries 1 and 3 of Figure 3a, replacing TfOH (entry 3) with AlBr₃ (entry 1) increases the ratio of 23:22 from 1.3 (42:32) to 8.5 (68:8). In my opinion, while maintaining a wavelength of 310 nm, AlBr₃ plays a significant role in selective control. Similarly, in entries 4 and 5 (with a wavelength of 390 nm), replacing TfOH (entry 5) with AlBr₃ (entry 4) increases the ratio of 21:22 from 0.9 (27:31) to 4.3 (64:15). Could the author explain why AlBr₃ enhances selectivity?

I think the way I designed Figure 3a might be misleading and the Reviewer is thinking that 22 is a reaction product while is the starting material. The use of AlBr₃ vs. TfOH does not alter the selectivity of the process but it just affects the conversion of *m*-cresol **2**₂ (starting material) to *p*-cresol **2**₃ at 310 nm and to *o*-cresol **2**₁ at 390 nm. In both cases, in presence of AlBr₃ we observe a better conversion to the desired product and additionally, at 390 nm the mass balance is also better. TfOH has to be used over-stoichiometric and this can lead to decomposition. I have changed Figure 3a replacing 22 with rsm.

Reviewer #4:
